# Functional Roles of *LaeA*-like Genes in Fungal Growth, Cellulase Activity, and Secondary Metabolism in *Pleurotus ostreatus*

**DOI:** 10.3390/jof8090902

**Published:** 2022-08-25

**Authors:** Guang Zhang, Peng Yan, Doudou Leng, Li Shang, Chaohui Zhang, Zhongwei Wu, Zhenhe Wang

**Affiliations:** Henan Institute of Science and Technology, College of Life Science and Technology, Xinxiang 453003, China

**Keywords:** LaeA, fungal growth, cellulase, polysaccharides, *Pleurotus ostreatus*

## Abstract

The global regulator LaeA plays crucial roles in morphological development and secondary metabolite biosynthesis in filamentous fungi. However, the functions of LaeA in basidiomycetes are less reported. The basidiomycete *Pleurotus ostreatus* is a well-known fungus used both in medicine and as food that produces polysaccharides and cellulolytic enzymes. In this study, we characterized three LaeA homologs (PoLaeA1, PoLaeA2, and PoLaeA3) in *P. ostreatus*. *PoLaeA1* showed different expression patterns than *PoLaeA2* and *PoLaeA3* during different developmental stages. Silencing *PoLaeA1* decreased the intracellular polysaccharide (IPS) content by approximately 28–30% and reduced intracellular ROS levels compared with those of the WT strain. However, silencing *PoLaeA2* and *PoLaeA3* decreased cellulase activity by 31–34% and 35–40%, respectively, and reduced the cytosolic Ca^2+^ content, compared with those of the WT strain. Further analysis showed that *PoLaeA1* regulated IPS biosynthesis through intracellular ROS levels, whereas *PoLaeA2* and *PoLaeA3* regulated cellulase activity through intracellular Ca^2+^ signaling. Our results provide new insights into the regulation of polysaccharide biosynthesis and cellulase production in filamentous fungi.

## 1. Introduction

The basidiomycete *Pleurotus ostreatus* is a well-known fungus used both in medicine and as food that has been cultivated and consumed worldwide. In recent years, the commercial cultivation of *P. ostreatus* is believed to be one of the fastest-growing sectors in the edible mushroom market [1]. On the one hand, *P. ostreatus* is commonly used as food or as a food ingredient for its rich nutrient content, including essential unsaturated fatty acids, vitamins, and amino acids [2]. On the other hand, *P. ostreatus* produces multiple bioactive compounds that possess physiological and therapeutic effects on human chronic disorders. Aqueous extracts of *P. ostreatus* have been reported to exhibit antioxidant and antihypertensive properties [3]. Among the bioactive compounds in *P. ostreatus*, polysaccharides are mainly responsible for their significant anti-inflammatory, antitumor, antidiabetic, antioxidant, and immunomodulatory roles [4]. Therefore, the polysaccharide content is becoming a measure of the quality of this fungus, and improving polysaccharide content is an important goal. In addition, *P. ostreatus* is an important white-rot fungus that produces cellulases and laccases to degrade cellulose and lignin, and this fungus has potential applications in energy production, bioconversion of agricultural wastes, and biodegradation of organo-pollutants and industrial contaminants [5]. Recently, *P. ostreatus* has been proven to improve biomethane production from rice straw via its cellulase and xylanase activities [6]. Understanding the regulatory mechanisms of cellulases and laccases is an urgent need and is helpful in environmental protection and energy deficiency relief efforts.

In recent years, *P. ostreatus* has become a potentially good model system for elucidating physiological and chemical processes in basidiomycete fungi, such as the generation of various bioactive compounds and a series of cellulose-degrading enzymes in a short production cycle [7]. A relatively well-annotated genome database of *P. ostreatus* has been published [8] and is available via GenBank (AYUK00000000), drawing interest in exploring this fungus. Moreover, genetic tools including gene silencing and gene overexpression systems [9] and efficient genetic transformation systems [10], have been applied in *P. ostreatus*, providing more possibilities to explore the functions of genes of interest.

Global regulators have been found to play crucial roles in morphological development and secondary metabolite biosynthesis and in other biological processes in filamentous fungi [11]. LaeA (loss of *aflR* expression A) is a key global regulator that was first identified genetically in *Aspergillus* spp. [12] and is a nuclear protein that is highly conserved in filamentous fungi. Deletion and overexpression of *LaeA* suggested that LaeA regulates the biosynthesis of sterigmatocystin, penicillin, and lovastatin through their respective metabolic gene clusters [12]. Subsequently, homologs of LaeA were found in other fungi, and studies have confirmed the important roles of LaeA homologs in regulating secondary metabolism and morphological developmental processes. In *Chaetomium globosum*, *CgLaeA* is required for pigmentation, sporulation, and chaetoglobosin A production [13]. In *Monascus purpureus*, overexpression of *LaeA* not only causes higher monacolin K and pigment production but also results in altered mycelial morphology [14]. In addition to its significant role in secondary metabolism, *LaeA* also contributes to the fungal virulence of *Aspergillus*
*ochraceus* [15] and the apple canker pathogen *Valsa mali* [16], among other fungi. In addition, *LaeA* was also found to regulate cellulolytic enzyme production in *Penicillium*
*oxalicum* [17] and *Trichoderma reesei* [18]. However, studies of the function of LaeA mainly focus on ascomycetes and are relatively rare in basidiomycete fungi due to a lack of effective genetic tools.

Here, three LaeA homologs (PoLaeA1, PoLaeA2, and PoLaeA3) were characterized in *P. ostreatus* to explore their regulatory role in secondary metabolism and cellulase activities. Silencing any of these *LaeA* homologous genes caused a defect in the normal growth of *P. ostreatus*. Further analysis showed that the intracellular ROS level was involved in the regulation of polysaccharide biosynthesis by *PoLaeA1*. However, *PoLaeA2* and *PoLaeA3* regulated cellulase activities via the cytosolic Ca^2+^ content.

## 2. Materials and Methods

### 2.1. Microbial Strains and Culture Conditions

The *P. ostreatus* strain (ACCC50596) obtained from the Agricultural Culture Collection of China was also preserved in the China Center for Mushroom Spawn Standards and Control (CCMSSC00389), and was named P89. The P89 strain was used as a parental strain to construct *PoLaeA*-silenced strains. The wild-type (WT), empty vector control (Sicontrol) and *LaeA*-silenced strains (LaeA1i-16, LaeA1i-38, LaeA2i-12, LaeA2i-26, LaeA3i-15 and LaeA3i-32) were grown on complete yeast medium (CYM: 2% glucose, 1% maltose, 0.2% tryptone, 0.2% yeast extract, 0.46% KH_2_PO_4_, 0.05% MgSO_4_·7H_2_O, 2% agar) at 28 °C [10]. Seed cultures of the fungal strains were prepared in potato dextrose broth at 28 °C and 150 rpm for 5 days. The cultures were broken up with a mechanical crusher under aseptic conditions, after which the cultures were transferred to 100 mL of CYM liquid medium. The fermentation experiments were performed in liquid CYM at 28 °C and 150 rpm for 7 days [19]. The *Escherichia coli* DH5α strain (TaKaRa, Dalian, China) cultured in lysogeny broth (LB) containing kanamycin (50 μg/mL) was used for plasmid construction. *Agrobacterium tumefaciens* GV3101 (IMCAS, Beijing, China) used in the transformation of *P. ostreatus* was grown in LB containing kanamycin (50 μg/mL) and rifampicin (50 μg/mL).

### 2.2. Gene Cloning and Sequence Analysis

To identify the homologs of LaeA in *P. ostreatus*, the sequence of LaeA from *Hypsizygus marmoreus* (GenBank accession number: RDB28437, accessed on 16 May 2022) was used in a homology search of the genome database of P89 [9]. Three *LaeA*-like genes (*PoLaeA1*, *PoLaeA2,* and *PoLaeA3* with the GenBank accession numbers ON804804, ON804805, and ON804806, respectively, accessed on 25 June 2022) were identified in the P89 genome database. The coding sequences of these genes were polymerase chain reaction (PCR) amplified using complementary DNA (cDNA) of P89 with the primers listed in Appendix A. The amplified fragments were cloned into the pMD19-T vector (TaKaRa, Dalian, China) for sequencing. The online tool ProtParam (http://web.expasy.org/protparam/, accessed on 16 May 2022) was used to predict the theoretical molecular weights and isoelectric points (pIs) of PoLaeA1, PoLaeA2, and PoLaeA3. The conserved domains present in the three LaeA-like proteins were analyzed using the NCBI Conserved Domains Database (http://www.ncbi.nlm.nih.gov/cdd, accessed on 16 May 2022). The phylogenetic tree was constructed with the MEGA 6.0 program [20] using the neighbor-joining method with 1000 bootstrap replicates.

### 2.3. Construction of PoLaeA-RNAi Plasmids and Strains

The original pBHG vector [21] was used to construct *PoLaeA*-silenced plasmids. Briefly, the sense and antisense fragments of the glyceraldehyde-3-phosphate dehydrogenase (gpd) promoter (GenBank: KY924471, accessed on 16 May 2022) of P89 were PCR amplified and digested and then inserted into pBHG at the corresponding restriction sites to generate the intermediate plasmid pHPG. The antisense fragments of *PoLaeA1*, *PoLaeA2,* and *PoLaeA3* were PCR amplified and doubly digested with *Bgl*Ⅱ and *Apa*I, followed by insertion into pHPG to generate the *PoLaeA*-silenced vectors RNAi-PoLaeA1, RNAi-PoLaeA2, and RNAi-PoLaeA3, respectively (Appendix A). Then, these constructed plasmids were transferred into *P. ostreatus* using *A. tumefaciens*-mediated transformation [10]. The transformants were randomly selected on CYM plates containing 90 μg/mL hygromycin B (hyg). In addition, the SiControl was constructed with pHPG without *PoLaeA* fragments. The silencing efficiency of these randomly selected *PoLaeA*-silenced strains randomly selected was analyzed by quantitative real-time PCR (qRT–PCR). The *PoLaeA-*silenced strains (LaeA1i-16, LaeA1i-38, LaeA2i-12, LaeA2i-26, LaeA3i-15, and LaeA3i-32) with higher silencing efficiency were selected and used for subsequent studies. The primers used are listed in Appendix A.

### 2.4. Measurement of Mycelial Growth and Biomass

The WT, SiControl and *PoL**aeA*-silenced strains were cultured on CYM plates at 28 °C for 7 days to detect the mycelial diameters of the resulting colonies. The relative mycelial diameters of the tested strains = (the mean diameter of the tested strains)/(the mean diameter of the WT strain) × 100%. Then, the dry weight (DW) of the fungal mycelia collected from CYM plates was measured gravimetrically after drying at 60 °C.

### 2.5. Measurement of the Intracellular Reactive Oxygen Species (ROS) Level

The fungal strains (WT, SiControl, and *PoL**aeA*-silenced strains) were cultured on CYM plates. Sterilized cover slips were obliquely inserted into the CYM plates adjacent to the fungal mycelia. After the fungal mycelia had grown onto the cover slips, the cover slips were stained with 10 μM 2′,7′-dichlorodihydrofluorescein diacetate (DCFH-DA) and then incubated at 28 °C for 30 min. The green fluorescence of H_2_O_2_ in fungal hyphae was measured with an LSM780 confocal laser scanning microscope (Zeiss, Jena, Germany) [19]. Fluorescence intensity was analyzed by ZEN 3.3 (blue edition). In addition, the fungal hyphae collected from liquid CYM cultures were used to measure the intracellular H_2_O_2_ content with a commercial hydrogen peroxide assay kit (Nanjing Jiancheng Bioengineering Institute, Nanjing, China) according to the instructions of the manufacturer [19]. The malondialdehyde (MDA) content of the fungal mycelia was determined as described previously [22]. Briefly, fungal mycelia were ground with liquid nitrogen, and then 0.5 mL of 5% trichloroacetic acid (TCA) was added. After centrifugation at 10,000× *g* for 10 min, the supernatant was collected and mixed with 0.5 mL of 0.67% β-thiobarbituric acid (TBA). The mixture was subsequently incubated at 95 °C for 10 min and then centrifuged at 10,000× *g* for 10 min. The absorbance of the supernatant was detected at 532 nm and corrected for nonspecific turbidity by subtracting the absorbance at 600 nm. The MDA content was calculated using the MDA extinction coefficient of 155 mM^−1^ cm^−1^ at 532 nm.

### 2.6. Enzymatic Activity Determination

The WT, SiControl and *PoL**aeA*-silenced strains were cultured in liquid CYM at 28 °C for 5 days, and the fungal hyphae were ground with liquid nitrogen, and suspended in 5 mL phosphate-buffered saline (PBS) buffer. After centrifugation at 12,000× *g* for 10 min at 4 °C, the supernatants were collected to measure the enzymatic activity of the antioxidant systems. The protein content was determined with the Bradford method with albumin from bovine serum as a standard. Ascorbate peroxidase (APX) activity was detected using a commercial ascorbate peroxidase assay kit (Nanjing jiancheng Bioengineering Institute, Nanjing, China). Catalase (CAT) activity was detected using a commercial catalase assay kit (Nanjing jiancheng Bioengineering Institute, Nanjing, China). Glutathione peroxidase (GPX) activity was measured by using a cellular glutathione peroxidase assay kit (Beyotime Institute of Biotechnology, Shanghai, China). Superoxide dismutase (SOD) were determined by monitoring the inhibition of the photochemical reduction of nitro blue tetrazolium [22].

To measure the endoglucanase (CMCase) activities of the fungal mycelia, the fungal strains were cultured in liquid CYM at 28 °C for 5 days, and the cultures were washed with 0.85% (*w*/*v*) NaCl, followed by transfer to MCM medium (1% cellulose, 0.05% MgSO_4_•7H_2_O, 0.5% (NH_4_)_2_SO_4_, 0.46% KH_2_PO_4_ and 2 mL/L trace element) with or without CaCl_2_ (5 mM) at 28 °C for 2 days. The culture supernatants were collected to detect the CMCase activity. A 2.0 mL reaction mixture (containing 0.5 mL diluted culture supernatants and 1.5 mL of 1% carboxymethylcellulose sodium salt) was transferred into a 25 mL tube and mixed followed by incubation at 50 °C for 30 min. 3 mL of DNS reagent were added to stop the reaction. A blank tube (with boiled crude enzyme) was used as a control to correct for the presence of reducing sugars in the crude enzyme samples [23]. The absorbance of the reaction mixture was detected at 540 nm with a spectrophotometer UV-1800 (Shimadzu Corporation, Kyoto, Japan). One unit of enzyme activity was defined as the amount of enzyme required to release 1 μmol of glycoside bonds of substrate per minute under defined assay conditions.

### 2.7. Measurement of the Cytosolic Ca^2+^ Content

Fluo-3AM (Invitrogen) is a membrane-permeable compound that can be hydrolyzed into Fluo-3 (a Ca^2+^-binding form) by endogenous esterase and was used as a fluorescent calcium indicator dye to measure the cytosolic Ca^2+^ level in fungal cells. The WT, SiControl and *PoLaeA*-silenced strains were cultured on CYM plates. Sterilized cover slips were obliquely inserted into the CYM plates adjacent to the fungal mycelia. After the fungal mycelia had grown onto the cover slips, the cover slips were stained with 50 μM Fluo-3AM and then incubated for 30 min. The fungal hyphae were viewed microscopically with an LSM780 confocal laser scanning microscope (Zeiss, Jena, Germany) [23].

### 2.8. Determination of Intracellular Polysaccharide Content

The WT, SiControl and *Po**LaeA*-silenced strains were cultured in liquid CYM at 28 °C for 7 days, and the fungal hyphae were collected. The intracellular polysaccharides (IPS) were extracted and analyzed as described previously [24]. Briefly, mycelia were washed with distilled water repeatedly and dried at 60 °C. Then the IPS of fungal mycelia was extracted with NaOH (1 M) and the supernatant was detected with the phenol-sulfuric acid method using glucose as a standard.

### 2.9. Analysis of Gene Expression with qRT–PCR

Synthesis of first-strand cDNA from the fungal strains (WT, SiControl, and *PoLaeA*-silenced strains) was performed with 5 × All-In-One RT MasterMix (ABM, Richmond, VA, Canada). qRT–PCR was performed with a Realplex2 System (Eppendorf, Hamburg, Germany) using EvaGreen 2 × qPCR MasterMix-S (ABM, Richmond, Canada) as described previously [25]. The glyceraldehyde-3-phosphate dehydrogenase gene (*GAPDH*) was used as a standard control. The 2^−^^△△CT^ method [26] was used to analyze the relative gene-specific mRNA expression levels. The primers used here are listed in Appendix A.

### 2.10. Data Analysis

Statistical analysis was carried out using GraphPad Prism Version 7.00 for Windows (GraphPad Software, San Diego, CA, USA, www.graphpad.com, accessed on 16 May 2022). All experiments were conducted in three biological replications with similar results. The mean values are interpreted as the mean ± standard deviation (SD). The error bars indicate the SD from the means of triplicates. One-way analysis of variance (ANOVA) followed by Dunnett’s multiple comparisons test was used to analyze the differences (*p* < 0.05) in the mean values between the analyzed samples.

## 3. Results

### 3.1. LaeA-like Genes in P. ostreatus

To identify LaeA homologs in *P. ostreatus*, LaeA of *Hypsizygus marmoreus* was used to BLAST against the genome database of P89. Using the criteria of E values of <1.0 × 10^−50^, three LaeA homologs were identified and termed *PoLaeA1*, *PoLaeA2*, and *PoLaeA3*. The open reading frames of *PoLaeA1*, *PoLaeA2*, and *PoLaeA3* are 1212, 756, and 699 bp, respectively, and are interrupted by six, six, and five introns, respectively. The *PoLaeA1* gene encodes a 45.75-kDa protein of 403 amino acids with a pI of 4.78; *PoLaeA2* encodes a 28.29-kDa protein of 251 amino acids with a pI of 5.44; and *PoLaeA3* encodes a 25.86 kDa protein of 232 amino acids with a pI of 4.32, respectively.

Domain analysis showed that all three LaeA-like proteins (PoLaeA1, PoLaeA2, and PoLaeA3) contain one Methyltransf_25 domain and belong to the AdoMet_Mtases superfamily, which uses S-adenosylmethionine (SAM) as a substrate for methyltransfer (Appendix A). In addition, these LaeA-like proteins revealed a reciprocal best BLAST hit to LaeA of *Hypsizygus marmoreus* and *Coprinopsis cinerea*, and alignments of these proteins demonstrated that structural homology with the SAM binding motif [12] is highly conserved (Figure 1). In addition, we investigated the divergence of the three LaeA-like proteins in *P. ostreatus* from other known LaeA proteins in filamentous fungi. The phylogenetic tree could be divided into two major groups (ascomycetes and basidiomycetes). The three LaeA-like proteins in *P. ostreatus* are most homologous to LaeA in *Hypsizygus marmoreus* and distinct from those of the ascomycetes (Appendix A).

### 3.2. Expression of LaeA-like Genes during Different Developmental Stages of P. ostreatus

The developmental stages of *P. ostreatus* include mycelia, primordia, morula, coral, forming, and mature stages (Appendix A), and we first examined the expression patterns of the *LaeA*-like genes during *P. ostreatus* development. The results showed that the expression of the three *LaeA*-like genes was significantly lower in mycelia than in any other stage. The expression of *PoLaeA1* was highest in the forming and mature stages and was 19.88- and 20.27-fold that of the mycelia (Figure 2A), respectively. However, the expression of *PoLaeA2* and *PoLaeA3* was highest in the morula and coral stages compared with expression in the mycelia (Figure 2B,C), respectively. In addition, we investigated the expression patterns of *LaeA*-like genes in different parts of the fruiting bodies of the WT strain. The expression of *PoLaeA1* was upregulated in the cap (1.95-fold), gill (2.42-fold), and pileipellis (2.13-fold) than in the stipe (Figure 2D). In contrast, the highest expression of *PoLaeA2* and *PoLaeA3* was in the stipe (Figure 2E,F). These results indicate that these *LaeA*-like genes are required for the development of *P. ostreatus* but might play different roles in development.

### 3.3. Construction of P. ostreatus LaeA-Silenced Strains

To investigate the roles of the *LaeA*-like genes in physiological processes, *PoLaeA*-silencing plasmids (RNAi-PoLaeA1, RNAi-PoLaeA2, and RNAi-PoLaeA3) were constructed using the pBHG vector, which contains the *Hyg* resistance gene as a selectable marker (Appendix A) and used to transform *P. ostreatus*. qRT–PCR analysis was used to measure the expression of *LaeA*-like genes and confirm the silencing efficiency in the randomly selected transformants. The expression of *PoLaeA1* in LaeA1i-16 and LaeA1i-38 was downregulated by 74% and 70%, and the expression of *PoLaeA2* in LaeA2i-12 and LaeA2i-26 was downregulated by 82% and 79%, and the expression of *PoLaeA3* in LaeA3i-15 and LaeA3i-32 was downregulated by 76% and 85%, respectively, compared with WT expression (Appendix A). Therefore, these *PoLaeA*-silenced strains were used for further study.

### 3.4. LaeA-like Genes Are Required for the Fungal Growth of P. ostreatus

To determine whether *LaeA*-like genes impact the growth of *P. ostreatus*, the WT, SiControl and *PoLaeA*-silenced strains were cultured on CYM plates to measure the resulting mycelial diameters. All *PoLaeA*-silenced strains grew much more slowly than the WT strain (Figure 3A). The mycelial diameters of LaeA1i-16, LaeA1i-38, LaeA2i-12, LaeA2i-26, LaeA3i-15, and LaeA3i-32 were approximately 23%, 20%, 28%, 16%, 37%, and 34% smaller, respectively, than that of the WT strain (Figure 3B). In addition, we measured the biomass of *PoLaeA*-silenced strains cultured in liquid CYM. The dry weights (DW) of *PoLaeA*-silenced strains were significantly lower, by approximately 21–44% than that of the WT strain (Figure 3C). These results together reveal that silencing *PoLaeA1*, *PoLaeA2* or *PoLaeA3* causes a defect in the normal growth of *P. ostreatus*.

### 3.5. Silencing PoLaeA1 Decreased the Intracellular ROS Levels of P. ostreatus

To decipher how *LaeA*-like genes affect fungal growth, we first examined intracellular ROS signaling in the *PoLaeA*-silenced strains using an H_2_O_2_ fluorescence probe (DCFH-DA). Figure 4A shows that the H_2_O_2_ fluorescence intensity of the *PoLaeA1*-silenced strains (LaeA1i-16 and LaeA1i-38) was significantly decreased, and the H_2_O_2_ fluorescence values in LaeAi-16 and LaeAi-38 were 40% and 41% that of the WT strain (Figure 4B), respectively. Furthermore, we tested the H_2_O_2_ levels of the *PoLaeA*-silenced strains. The intracellular H_2_O_2_ levels in LaeA1i-16 and LaeA1i-38 were 11.37 and 12.01 mM/g protein and were reduced by 42% and 39% (Figure 4C), respectively, compared with that of the WT strain. The MDA content can reflect the status of intracellular ROS [22]. In addition, the MDA content in LaeA1i-16 and LaeA1i-38 was reduced by approximately 45% and 49%, respectively, compared with that of the WT strain (Appendix A). However, the H_2_O_2_ fluorescence intensity, the intracellular H_2_O_2_ levels, and the MDA content in the *PoLaeA2*-silenced strains (LaeA2i-12 and LaeA2i-26) and *PoLaeA3*-silenced strains (LaeA3i-15 and LaeA3i-32) did not differ from those of the WT strain (Figure 4 and Appendix A). These results indicate that silencing *PoLaeA1* reduces intracellular ROS levels in *P. ostreatus*.

Next, we examined the major enzymes involved in the antioxidant system to understand how *PoLaeA1* affects intracellular ROS levels. Figure 5 shows that the activities of APX (catalyzes the oxidation of ascorbic acid), CAT (decomposes H_2_O_2_), GPX (transfers glutathione to glutathione disulfide), and SOD (superoxide dismutase) in LaeA1i-16 and LaeA1i-38 were approximately 57–58%, 56–68%, 76–78%, and 64–65% higher, respectively, than those of the WT strain. Meanwhile, the expression of the genes *APX*, *CAT1*, *CAT2*, *GPX*, *SOD1*, and *SOD2* in LaeA1i-16 and LaeA1i-38 was significantly upregulated compared with expression in the WT strain (Appendix A). These results imply that silencing *PoLaeA1* increases the activity of enzymes involved in the antioxidant system.

### 3.6. Silencing PoLaeA2 and PoLaeA3 Reduced the Cytosolic Ca^2+^ Content of P. ostreatus

Ca^2+^ signaling plays a crucial role in fungal growth, and we measured the cytosolic Ca^2+^ content in *PoLaeA*-silenced strains using the Ca^2+^ fluorescence probe Fluo-3AM. As shown in Figure 6A, the cytosolic Ca^2+^ content was lower in the *PoLaeA2*- and *PoLaeA3*-silenced strains than in the WT strain. Fluorescence analysis showed that the Ca^2+^ fluorescence values in *PoLaeA2*- or *PoLaeA3*-silenced strains were approximately 67–78% lower than that of the WT strain (Figure 6B). However, the cytosolic Ca^2+^ content in the *PoLaeA1*-silenced strains was not different from that of the WT strain. Moreover, we examined the transcriptional regulation of Ca^2+^ signaling in the *PoLaeA2*- and *PoLaeA3*-silenced strains, including Ca^2 +^ -permeable channel (Cch and Mid), vacuole Ca^2+^ channel (Yvc), phospholipase C (Plc), calmodulin (Cam), calcineurin (Cna1 and Cna2), calcineurin-responsive zinc finger transcription factor (Crz), Ca^2+^/Cam-dependent protein kinase (Camk1, Camk2 and Camk3), and calreticulin and Ca^2+^ binding protein (CABP). With the exceptions of *Cna2*, *Camk2,* and *Camk3*, the expression of Ca^2+^ signaling genes in *PoLaeA2*- and *PoLaeA3*-silenced strains was significantly downregulated compared with WT expression (Figure 6C). These results suggest that silencing *PoLaeA2* and *PoLaeA3* decreased the cytosolic Ca^2+^ content of *P. ostreatus.*

### 3.7. The IPS Contents Were Decreased in PoLaeA1-Silenced Strains and Could Be Recovered by H_2_O_2_

To test whether the *LaeA*-like genes affect the metabolites of *P. ostreatus*, the IPS contents of the *PoLaeA*-silenced strains were determined. The IPS contents in LaeA1i-16 and LaeA1i-38 were 68.95 and 66.94 mg/g DW and were approximately 28% and 30% lower than that of the WT strain (Figure 7A), respectively. However, the IPS contents of the *PoLaeA2*- and *PoLaeA3*-silenced strains were not different from that of the WT strain. Meanwhile, the transcripts of three key genes involved in polysaccharide biosynthesis, namely, *Ugp* (encodes UDP-glucose pyrophosphorylase), *Pgm* (encodes phosphoglucomutase), and *Pgi* (encodes glucose-6-phosphate isomerase), were all significantly lower in the *PoLaeA1*-silenced strains than in the WT strain (Figure 7B). The decreased IPS contents and reduced ROS content in the *PoLaeA1*-silenced strains might be related; to test this hypothesis, we added 1 mM H_2_O_2_ to the *PoLaeA1*-silenced strains. Figure 7C shows that adding H_2_O_2_ restored the decreased IPS contents in the *PoLaeA1*-silenced strains to the levels found in the WT strain. In addition, the ROS scavengers N-acetylcysteine (NAC) and ascorbate (Vc) were used to confirm the effect of H_2_O_2_ on the restored IPS contents in *PoLaeA1*-silenced strains. As expected, the IPS contents in the *PoLaeA1*-silenced strains decreased again after treatment with H_2_O_2_ plus 1 mM NAC or 1 mM Vc (Figure 7C). In addition, the downregulated expression of *Ugp* in *PoLaeA1*-silenced strains was restored to that of the WT strain when H_2_O_2_ was added and decreased again after treatment with H_2_O_2_ plus NAC or Vc (Figure 7D), presenting the same trend as the IPS contents. These results together indicate that the intracellular ROS level is involved in polysaccharide biosynthesis regulated by *PoLaeA1.*

### 3.8. The Cytosolic Ca^2+^ Content Affected the Regulation of Cellulase Activities by PoLaeA2 and PoLaeA3

Cellulase activity is important for fungal growth; therefore, we measured endoglucanase (CMCase) activity to investigate the influence of *LaeA*-like genes on cellulase production in *P. ostreatus*. Figure 8A shows significantly lower CMCase activities in the *PoLaeA2*- and *PoLaeA3*-silenced strains, approximately 31–34% and 35–40% lower than that of the WT strain, respectively. Furthermore, the expression of genes related to cellulase (*Cbh1*, *Cbh2*, *Cbh3,* and *Cbh4*, encode cellobiohydrolases; *Eg1*, *Eg2*, *Eg3,* and *Eg4*, encode endoglucanases; *Bgl1* and *Bgl2*, encode beta-glucosidases) was examined. The expression of three of the *Cbh* genes (*Cbh1*, *Cbh2,* and *Cbh3*), three of the *Eg* genes (*Eg1*, *Eg2,* and *Eg3*) and one of the *Bgl* genes (*Bgl1*) was significantly decreased in the *PoLaeA2*- or *PoLaeA3*-silenced strains compared with that in the WT strain (Figure 8B). These results indicate that silencing *PoLaeA2* and *PoLaeA3* causes a reduction in cellulase activity in *P. ostreatus.*

We further examined the relationship between the decreased Ca^2+^ content and the decreased cellulase activity in the *PoLaeA2*- and *PoLaeA3*-silenced strains. Figure 8C shows that adding 5 mM CaCl_2_ recovered the decreased CMCase activities in the *PoLaeA2*- and *PoLaeA3*-silenced strains to the levels found in the WT strain. Moreover, two Ca^2+^ inhibitors (LaCl_3_ and EGTA) were used to confirm the effect of Ca^2+^ content on the CMCase activities in the *PoLaeA2*- and *PoLaeA3*-silenced strains. As expected, the CMCase activities in the *PoLaeA2*- and *PoLaeA3*-silenced strains were decreased again after treatment with CaCl_2_ plus 5 mM EGTA or 5 mM LaCl_3_ (Figure 8C). In addition, the downregulated expression of *EG1* in the *PoLaeA2-* and *PoLaeA3*-silenced strains was restored to that of the WT strain when CaCl_2_ was added and decreased again after treatment with CaCl_2_ plus EGTA or LaCl_3_ (Figure 8D). Together, these results imply that *PoLaeA2* and *PoLaeA3* improved cellulase activities through Ca^2+^ signaling.

## 4. Discussion

LaeA is a global regulator that plays a crucial role in secondary metabolite biosynthesis in filamentous fungi [27]. Since its initial identification in *Aspergillus nidulans* [12], numerous orthologs of LaeA have been identified and found to be highly conserved in filamentous fungi. LaeA and its orthologs contain a SAM binding domain and have some similarities to methyltransferases [28]. Numerous studies have revealed that LaeA orthologs are important for secondary metabolism biosynthesis and morphological development and for differentiation in filamentous fungi [11]. However, the functions of LaeA and its orthologs in edible fungi are relatively unknown. In the present study, we identified and characterized three LaeA homologs (PoLaeA1, PoLaeA2, and PoLaeA3) in *P. ostreatus*. Despite their similar roles in fungal growth, *PoLaeA1* has different roles in secondary metabolism and cellulase production than *PoLaeA2* and *PoLaeA3* in *P. ostreatus.* Similarly, the *LaeA*-like genes (*LaeA*, *LaeA2,* and *LaeA3*) play different roles in citric acid accumulation in *Aspergillus luchuensis* mut. *kawachii* [29].

In ascomycetes, LaeA and its orthologs have a minor effect on fungal growth but are reported to participate in morphological and fungal development. The developmental regulatory role of LaeA has been well established in ascomycetes. In *A**spergillus nidulans*, *LaeA* is reported to control both sexual and asexual development [30] and is involved in light-responding developmental regulation [31]. In the present study, all three *LaeA* homologs were expressed at lower levels in mycelia than in any other developmental stage, implying that they are also required for fungal development. In addition, *PoLaeA1* showed different expression patterns than *PoLaeA2* and *PoLaeA3* during the developmental stages of *P. ostreatus* (Figure 3), indicating that *PoLaeA1* might play different roles than *PoLaeA2* and *PoLaeA3*. Furthermore, we found that all three *LaeA* homologs were required for normal growth in *P. ostreatus*. Similarly, loss of *LaeA* decreases the colony diameters in *A**spergillus ochraceus* [15]. Nevertheless, given the few reports of *LaeA* on fungal growth in ascomycetes, its role in the regulation of fungal growth in basidiomycetes might differ from that of *LaeA* in ascomycetes.

As a global regulator, LaeA has been well characterized in secondary metabolism in numerous ascomycete fungi but in a few basidiomycetes. LaeA is reported to regulate secondary metabolism by forming velvet complexes to induce chromatin modification [32]. To date, the regulatory roles of LaeA on secondary metabolism-related genes, such as sterigmatocystin [12] and dothistromin [33] have been extensively studied. In addition, LaeA was also found to regulate the biosynthesis of organic acids. Knockout of *LaeA* eliminated kojic acid biosynthesis in *A**spergillus oryzae* [34] and inhibited citric acid production in *A**spergillus luchuensis* mut. *kawachii* [29]. Consistently, silencing *PoLaeA1* decreased IPS production in *P. ostreatus*. In addition, silencing *PoLaeA1* also decreased the intracellular ROS levels of *P. ostreatus*. Our further study showed that *PoLaeA1* regulated IPS production through intracellular ROS signaling. Consistent with our results, intracellular ROS signaling is also involved in the regulation of secondary metabolism in *Ganoderma lucidum* [35]. However, the knockout of *LaeA* in *Coprinopsis cinerea* unexpectedly upregulates the biosynthesis of coprinoferrin [36]. These findings indicate that the role of LaeA in the regulation of secondary metabolism seems to be species specific in basidiomycetes.

Many filamentous fungi secrete cellulolytic enzymes, including cellulases, that are helpful in biofuel refining. Recently, LaeA was also found to regulate cellulase production in several ascomycetes. In *Penicillium oxalicum*, *LaeA* is reported to positively regulate prominent cellulase and hemicellulase but negatively regulate β-xylosidase formation [17]. In *Trichoderma reesei*, the deletion of *LaeA* completely blocks the expression of all seven cellulases, β-glucosidases, and xylanases [18]. As expected, we found that silencing *PoLaeA2* and *PoLaeA3* decreased cellulase production and downregulated the expression of cellulase-related genes. The signals involved in the regulation of cellulase production are limited thus far. The Ca^2+^-responsive signaling pathway is reported to play a crucial role in regulating cellulase production in *Trichoderma reesei* [37]. In addition, Ca^2+^ signaling is also involved in the regulation of cellulase by MAPK signaling [23]. These studies suggest that Ca^2+^ signaling is essential for cellulase production in filamentous fungi. However, the relationship between Ca^2+^ signaling and LaeA has not been reported. Our study showed that *PoLaeA2* and *PoLaeA3* positively regulated cellulase activities in Ca^2+^ signaling. Moreover, we found that *PoLaeA2* and *PoLaeA3* positively regulated the cytosolic Ca^2+^ content in *P. ostreatus*. In addition, silencing *PoLaeA2* and *PoLaeA3* downregulated the expression of Ca^2+^ signaling genes, demonstrating that *PoLaeA2* and *PoLaeA3* could positively regulate the influx of Ca^2+^. In addition, our pharmacological experiments indicated that the cytosolic Ca^2+^ content promotes the regulation of cellulase activities by *PoLaeA2* and *PoLaeA3*. However, the specific regulatory mechanism of *PoLaeA2* and *PoLaeA3* on cellulase is still unclear and should be clarified in the future. Nevertheless, our study provides new insights into the regulation of cellulase production in filamentous fungi.

In summary, we characterized three LaeA homologs (PoLaeA1, PoLaeA2, and PoLaeA3) in *P. ostreatus*. These LaeA orthologs are all required for the normal growth of *P. ostreatus. PoLaeA1* plays different roles than *PoLaeA2* and *PoLaeA3* in *P. ostreatus*. *PoLaeA1* regulates IPS content through intracellular ROS levels, whereas *PoLaeA2* and *PoLaeA3* regulate cellulase activity via cytosolic Ca^2+^ signaling. Our findings provide new insights into the regulation of polysaccharide biosynthesis and cellulase production in filamentous fungi.

## Figures and Tables

**Figure 1 jof-08-00902-f001:**
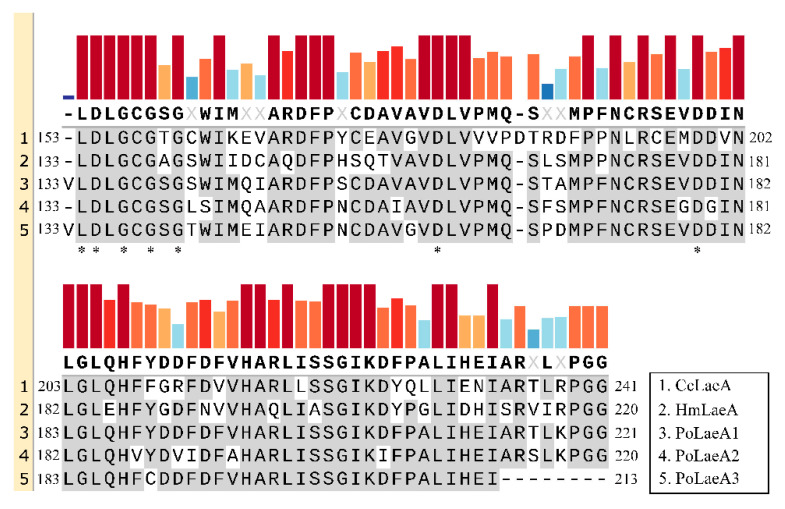
Amino acid sequence alignments of the conserved domain in LaeA-like proteins from basidiomycetes. The conserved domains were identified by Pfam (https://pfam.xfam.org/, accessed on 16 May 2022). The GenBank accession numbers for the LaeA-like proteins are *Coprinopsis cinerea* CcLaeA (XP_001829319, accessed on 16 May 2022) and *Hypsizygus marmoreus* HmLaeA (RDB28437, accessed on 16 May 2022). The results were constructed with SnapGene software (version 6.0.5) using the multiple sequence comparison function. Asterisks indicate the putative S-adenosylmethionine (SAM) binding sites in the conserved domain.

**Figure 2 jof-08-00902-f002:**
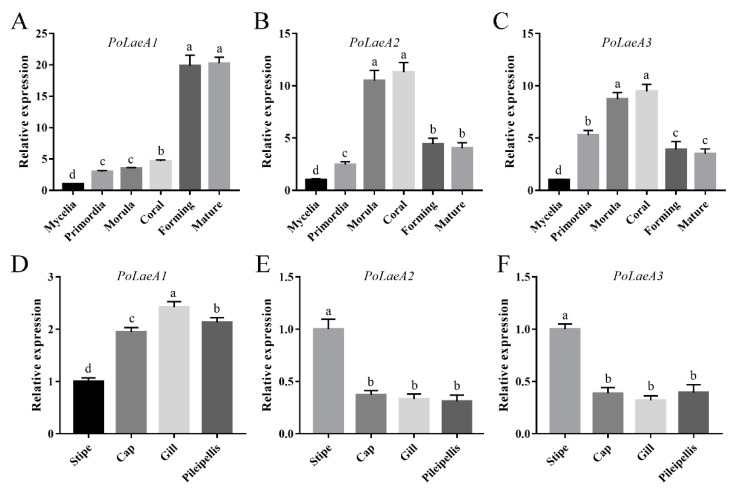
The expression of *LaeA*-like genes during different developmental stages and in different parts of fruiting bodies. (**A**–**C**) qRT–PCR analysis of *LaeA*-like gene expression in the WT strain during different developmental stages. The relative abundances of *PoLaeA1* (**A**), *PoLaeA2* (**B**) and *PoLaeA3* (**C**) transcripts at different stages were normalized by comparison with those levels in the mycelial stage (relative transcript level = 1.0). (**D**–**F**) Expression of *LaeA*-like genes in different parts of fruiting bodies. The relative abundances of *PoLaeA1* (**D**), *PoLaeA2* (**E**) and *PoLaeA3* (**F**) transcripts in different parts of fruiting bodies were normalized by comparison with those levels in the stipe (relative transcript level = 1.0). The values are the mean ± SD (*n* = 3). Different lowercase letters indicate significant differences between the analyzed samples (*p* < 0.05, according to Dunnett’s multiple comparisons test).

**Figure 3 jof-08-00902-f003:**
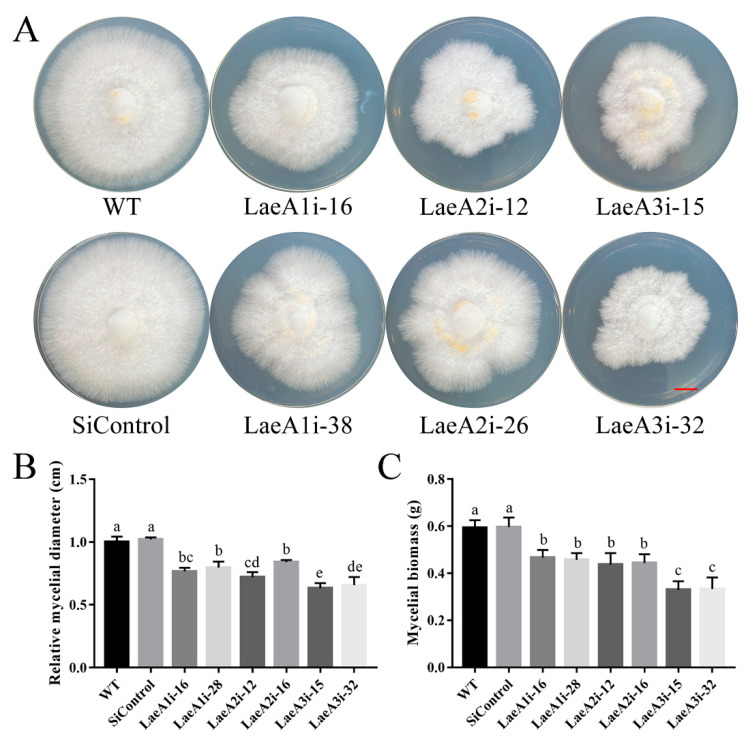
Hyphal growth and biomass of *PoLaeA*-silenced strains. (**A**) Fungal growth of *P. ostreatus* strains grown on CYM plates at 28 °C for 7 days. (Red scale bar = 1 cm). (**B**) Mycelial diameters. (**C**) Mycelial biomass. The values are the mean ± SD (*n* = 3). Different lowercase letters indicate significant differences between the strains (*p* < 0.05, according to Dunnett’s multiple comparisons test).

**Figure 4 jof-08-00902-f004:**
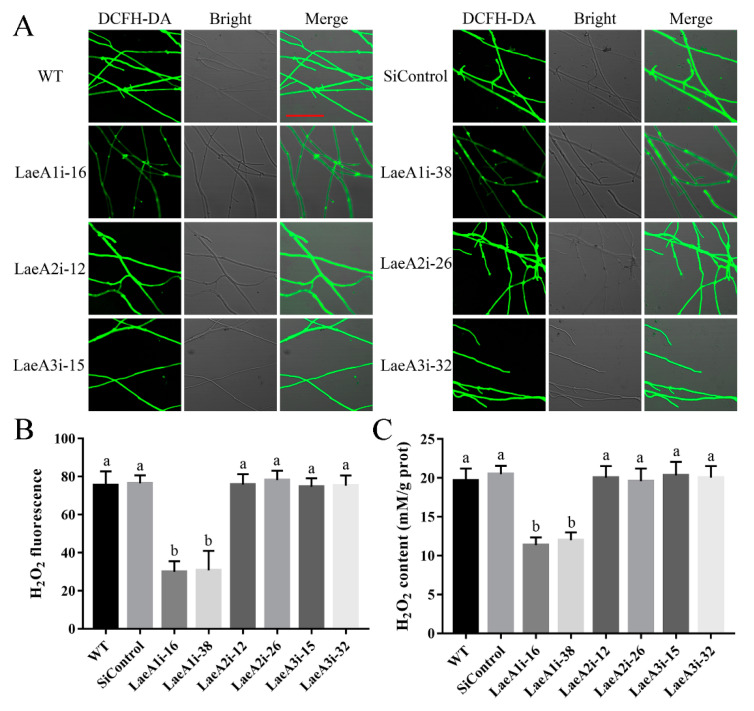
Silencing *PoLaeA1* decreased the intracellular ROS levels. (**A**) DCFH-DA staining (scale bar = 100 μm). The tested strains cultured on CYM plates were stained with the fluorescent H_2_O_2_ probe DCFH-DA. (**B**) The H_2_O_2_ fluorescence values. The fluorescence was analyzed with ZEN 3.3 (blue edition). (**C**) Intracellular H_2_O_2_ levels. The tested strains were cultured in liquid CYM at 28 °C for 5 days. The grounded mycelia were used to determine the intracellular H_2_O_2_ levels with a commercial hydrogen peroxide assay kit (Nanjing Jiancheng Bioengineering Institute, Nanjing, China). The values are the mean ± SD (*n* = 3). Different lowercase letters indicate significant differences between the strains (*p* < 0.05, according to Dunnett’s multiple comparisons test).

**Figure 5 jof-08-00902-f005:**
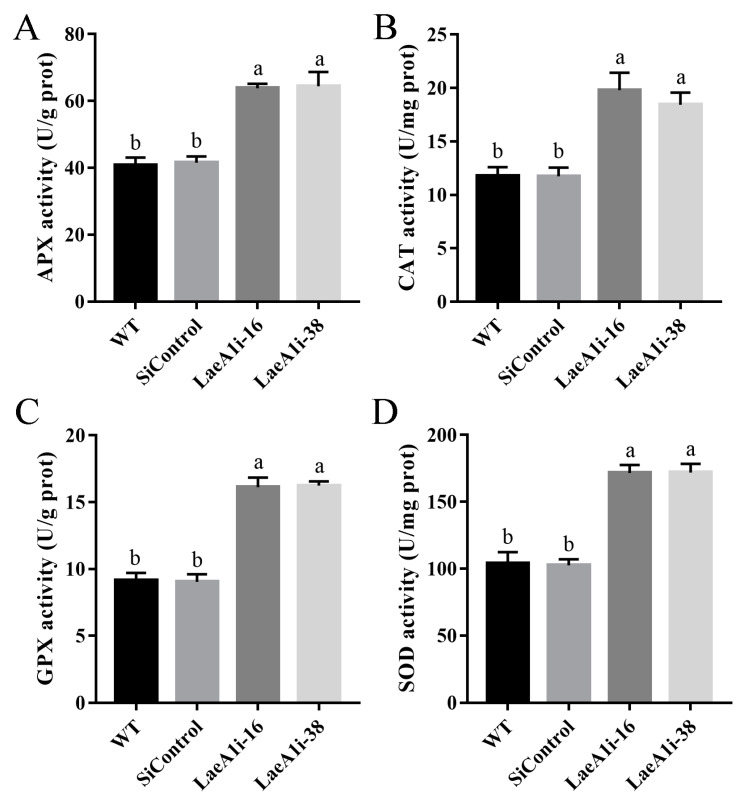
The activities of enzymes involved in the antioxidant system in *PoLaeA*1-silenced strains. The *P. ostreatus* strains (WT, SiControl, LaeA1i-16 and LaeA1i-38) were cultured in liquid CYM at 28 °C for 5 days. The grounded mycelia were suspended in phosphate-buffered saline buffer. After centrifuging, the supernatants were used to detect the enzyme activities involved in ROS scavenging. (**A**) APX activity. (**B**) CAT activity. (**C**) GPX activity. (**D**) SOD activity. The values are the mean ± SD (*n* = 3). Different lowercase letters indicate significant differences between the strains (*p* < 0.05, according to Dunnett’s multiple comparisons test).

**Figure 6 jof-08-00902-f006:**
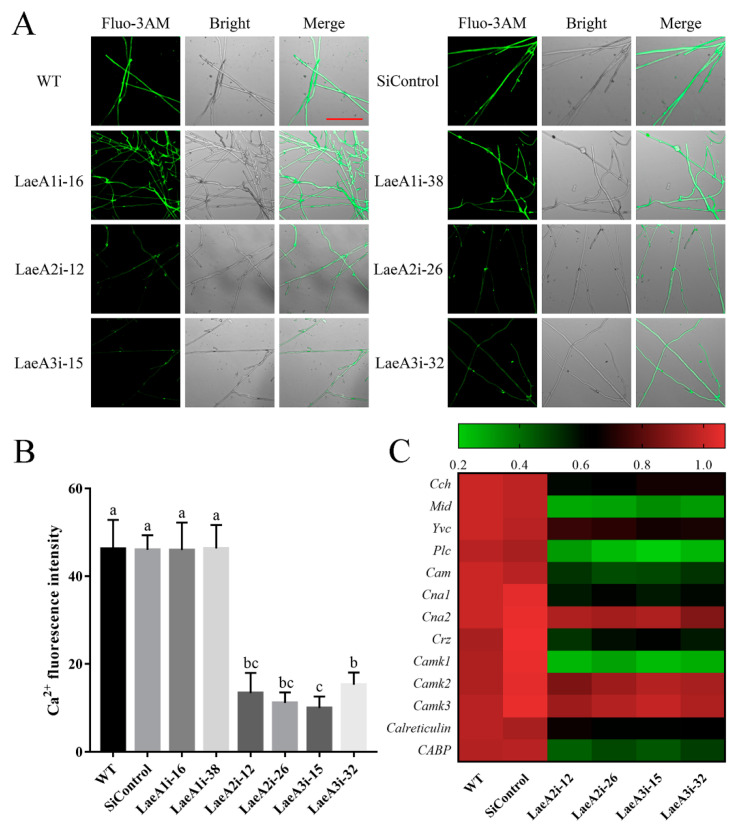
The cytosolic Ca^2+^ content in the *PoLaeA*-silenced strains. The fungal strains were cultured on CYM plates at 28 °C for 4 days. (**A**) Fluo-3AM staining (scale bar = 100 μm). For detection, 50 μM Fluo-3AM, a Ca^2+^ fluorescent probe, was used to monitor the fluorescence intensity using an LSM780 confocal laser scanning microscope. Green fluorescence represents the free cytosolic Ca^2+^. (**B**) Ca^2+^ fluorescence values. The fluorescence was analyzed with ZEN 3.3 (blue edition). (**C**) Transcriptional analysis of Ca^2+^-related genes. The expression levels of the Ca^2+^-related genes in the WT strain were arbitrarily set to 1.0. The values are the mean ± SD (*n* = 3). Different lowercase letters indicate significant differences between the strains (*p* < 0.05, according to Dunnett’s multiple comparisons test).

**Figure 7 jof-08-00902-f007:**
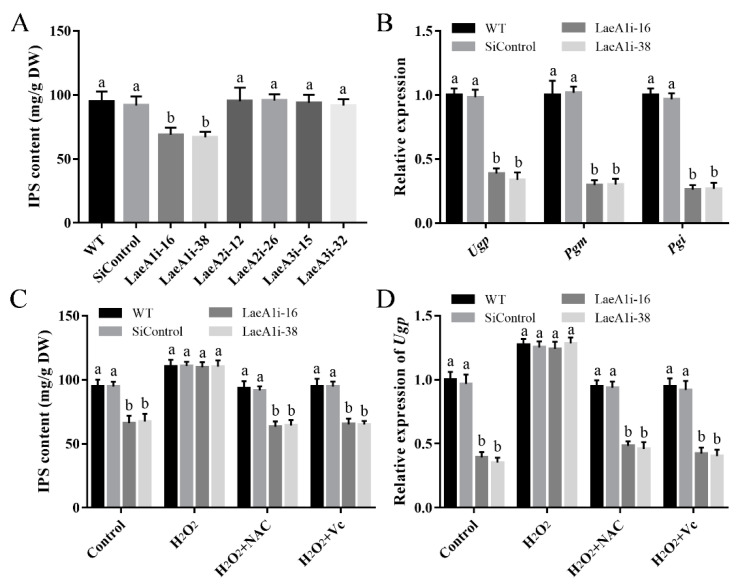
The intracellular ROS level was involved in the regulation of intracellular polysaccharide (IPS) content by *PoLaeA1*. The *P. ostreatus* strains were cultured in liquid CYM treatment with or without 1 mM H_2_O_2_, 1 mM NAC or 1 mM Vc at 28 °C for 7 days. (**A**) IPS content of the *PoLaeA*-silenced strains. (**B**) Transcriptional analysis of polysaccharide biosynthesis-related genes. (**C**) H_2_O_2_ recovered the decreased IPS content in the *PoLaeA1*-silenced strains. The IPS contents in the fungal mycelia were measured with the phenol-sulfuric acid method using glucose as a standard. (**D**) Transcriptional analysis of *Ugp*. The expression level of the *Ugp* gene in the WT strain of the control was arbitrarily set to 1.0. The values are the mean ± SD (*n* = 3). Different lowercase letters indicate significant differences between the strains (*p* < 0.05, according to Dunnett’s multiple comparisons test).

**Figure 8 jof-08-00902-f008:**
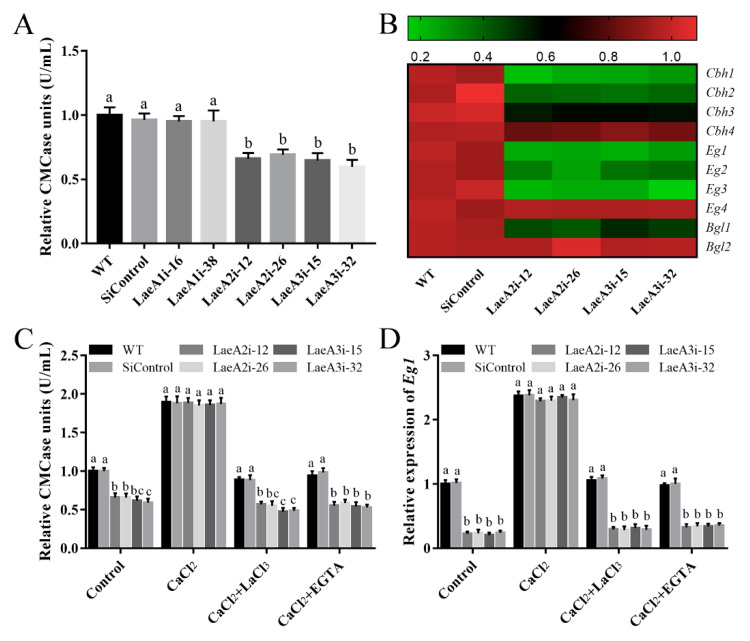
The cytosolic Ca^2+^ content was involved in the regulation of cellulase activity by *PoLaeA2* and *PoLaeA3*. The *P. ostreatus* strains cultured in liquid CYM at 28 °C for 5 days were transferred into MCM using cellulose as the sole carbon source, and treated with 5 mM CaCl_2_, LaCl_3_ or EGTA for 24 h. (**A**) CMCase (endoglucanase) activities of the *PoLaeA*-silenced strains. (**B**) Relative expression of the cellulase genes. The expression levels of the cellulase genes in the WT strain were arbitrarily set to 1.0. (**C**) CaCl_2_ recovered the decreased CMCase activities in the *PoLaeA2*- and *PoLaeA3*-silenced strains. (**D**) Transcriptional analysis of *Eg1*. The values are the mean ± SD (*n* = 3). Different lowercase letters indicate significant differences between the strains (*p* < 0.05, according to Dunnett’s multiple comparisons test).

## Data Availability

Not applicable.

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
