# Peer review of "Functional Roles of LaeA-like Genes in Fungal Growth, Cellulase Activity, and Secondary Metabolism in Pleurotus ostreatus"

_jof, 2022, doi:10.3390/jof8090902_

Round 1

Reviewer 1 Report

In the manuscript by Zhang et al., the authors characterize three LaeA homologs in basidiomycete fungus Pleurotus ostreatus. As acknowledged by the authors, LaeA is an important regulator of fungal morphology and development in Aspergillus spp. Its characterization in P. ostreatus provides an insight into LaeA’s role in basidiomycetes. The authors study the LaeA function in P. ostreatus by generating knockdown strains of PoLaeA1, PoLaeA2, and PoLaeA3. The authors achieved appreciable levels of knockdowns for each gene and the study is robust with the characterization of at least two knockdown strains for each gene. The experiments performed are appropriate and the results show the involvement of PoLaeA1 in the IPS production dependent on ROS levels and PoLaeA2 and PoLaeA3 in the Ca2+-mediated cellulase activity. One minor experimental result that was difficult appreciate is shown in Fig. S3. While there might be differences in the branch lengths of areal hyphae of wt and LaeA mutants, it is difficult to appreciate this difference through the pictures presented. The quantification results are convincing; however, it is unclear whether these results add to the growth differences shown in Fig.3. In addition, some of the results sub-sections can be described more clearly to improve the understanding of the reader. Minor points are listed below:

-       Figure legends for the supplementary figures are missing.

-       In the paragraph starting with Line 50, when LaeA is referred to as protein, it should not be italicized.

-       Materials & methods: Authors often refer to the methods as “previously described” and provide a citation. While it is important to cite previous literature, the brief description of the method in the present study would enhance the readers understanding of the results.

-       The paragraph starting with line 187 should be moved to materials and methods, perhaps under “data analysis”. The current data analysis can be relabeled as “Bioinformatic analysis”.

-       Line 192, results section can start from here but an introductory sentence is needed to remind the reader the motivation of the study.

-       Figure 2, it would be nice to have a diagram showing the developmental phases and structures of Pleurotus ostreatus.

-       If Fig.S3 is taken out, lines 257-261 and lines 417-418 need to be taken out.

-       In line 278, brief introductory sentence explaining why the authors looked at MDA content needs to be included.

-       In figure legends, some description of the method used should be included. For example in Figure 4, it is just mentioned that intracellular H2O2 levels were measured, but how they are measured or which type of cells were used unclear. In addition, they values of a,b,c,bc, etc. for the P-values should be explicitly stated.

Author Response

Dear Editors and reviewers:

Thank you for your letter and for the reviewers′ precious comments and suggestions concerning our manuscript entitled “Functional roles of LaeA-like genes in fungal growth, cellulase activity, and secondary metabolism in Pleurotus ostreatus” (ID is jof-1856782). These comments were all valuable and very helpful for revising and improving our paper and significantly guided our research. We have carefully revised the manuscript according to your comments. In addition, thank you for the guidelines on how to submit our revised manuscript. We hope the revised manuscript is now suitable for publication.

The sections revised according to the reviewer′s suggestions are highlighted in red text in the revised manuscript. The point-to-point replies to the reviewers′ comments are listed below.

Reviewers' comments:

Reviewer #1:

In the manuscript by Zhang et al., the authors characterize three LaeA homologs in basidiomycete fungus Pleurotus ostreatus. As acknowledged by the authors, LaeA is an important regulator of fungal morphology and development in Aspergillus spp. Its characterization in P. ostreatus provides an insight into LaeA’s role in basidiomycetes. The authors study the LaeA function in P. ostreatus by generating knockdown strains of PoLaeA1, PoLaeA2, and PoLaeA3. The authors achieved appreciable levels of knockdowns for each gene and the study is robust with the characterization of at least two knockdown strains for each gene. The experiments performed are appropriate and the results show the involvement of PoLaeA1 in the IPS production dependent on ROS levels and PoLaeA2 and PoLaeA3 in the Ca2+-mediated cellulase activity. One minor experimental result that was difficult appreciate is shown in Fig. S3. While there might be differences in the branch lengths of areal hyphae of wt and LaeA mutants, it is difficult to appreciate this difference through the pictures presented. The quantification results are convincing; however, it is unclear whether these results add to the growth differences shown in Fig.3.

Response: Thank you very much for your comments. We added “Fig. S3” here to show the difference of hyphal branches of WT and LaeA mutants. However, there is less relations between “Fig. 3” and “Fig. S3”, and this will confuse the readers. To make readers clearer, we have deleted the “Fig. S3” and the relative description in our revised manuscript based on your suggestion. Thank you again.

In addition, some of the results sub-sections can be described more clearly to improve the understanding of the reader. Minor points are listed below:

  1. Figure legends for the supplementary figures are missing

Response: Thank you very much. We have added figure legends for the Supplementary Figures in the Section of “Supplementary Materials”.

  1. In the paragraph starting with Line 50, when LaeA is referred to as protein, it should not be italicized.

Response: Thank you very much for your suggestion. We have changed the font of LaeA that is referred to as protein according to your suggestions (Page 2, line 58, 60, 61 and 70).

  1. Materials & methods: Authors often refer to the methods as “previously described” and provide a citation. While it is important to cite previous literature, the brief description of the method in the present study would enhance the readers understanding of the results.

Response: Thank you very much for your suggestion. We have added “the brief description of the method” while it is important to cite previous literature in the Section of “Materials and methods” (Page 3, lines 141-142 and 148-149; Page 4, lines 156-159, 166-168 and 172-174).

  1. The paragraph starting with line 187 should be moved to materials and methods, perhaps under “data analysis”. The current data analysis can be relabeled as “Bioinformatic analysis”.

Response: Thank you very much. We have moved “line 187-191” to the Section of the “Data analysis” of “Materials and methods” accordingly (Page 4, line 184-190). In addition, we moved the current data analysis to the Section of “3.1 LaeA-like genes in P. ostreatus” of the “Results” accordingly (Page 4, lines 193-196; Page 5, lines 197-200).

  1. Line 192, results section can start from here but an introductory sentence is needed to remind the reader the motivation of the study.
    Response: Thank you very much for your suggestion. We moved the line 177-184 to the Section of “3.1 LaeA-like genes in P. ostreatus” of the “Results” according to your fourth suggestion (Page 4, lines 193-196; Page 5, lines 197-200) and this will remind the reader the motivation of the study.

  1. Figure 2, it would be nice to have a diagram showing the developmental phases and structures of Pleurotus ostreatus.

Response: Thank you very much for your suggestion. We added a diagram to show the developmental phases and structures of Pleurotus ostreatus in “Figure S3” (Page 5, line 223) of the Section of “Supplementary materials”.

  1. If Fig.S3 is taken out, lines 257-261 and lines 417-418 need to be taken out.

Response: Thank you very much. We have taken out the Fig.S3 here. Besides, the relative description (lines 257-261 and lines 417-418) have been taken out accordingly.

  1. In line 278, brief introductory sentence explaining why the authors looked at MDA content needs to be included.

Response: Thank you very much for your suggestion. We have added the sentence “The MDA content can reflect the status of intracellular ROS” to explain why we looked at MDA content here (Page 7, line 284).

  1. In figure legends, some description of the method used should be included. For example in Figure 4, it is just mentioned that intracellular H2O2 levels were measured, but how they are measured or which type of cells were used unclear. In addition, they values of a,b,c,bc, etc. for the P-values should be explicitly stated.

Response: Thank you very much for your suggestion. We have added the description of the method used in the “Figure lengends” accordingly (Page 7, line 272; Page 8, lines 303 and 305-308; Page 9, lines 314-316; Page 10, lines 338-343; Page 11, lines 374-376; Page 12, lines 410-414). We also added the method for the detection of H2O2 levels in Figure 4. In addition, we have added the method used for P-values in the figure legends accordingly (Page 6, lines 245-246; Page 7, line 274; Page 8, lines 309; Page 9, line 318; Page 10, lines 344-345; Page 11, line 378; Page 12, lines 413-414).

Thanks again for your comments and suggestions.

Best wishes,

Guang Zhang

13 Aug 2022

Reviewer 2 Report

1.  In the results, there is a section on the measurement of growth and biomass.  However, there is no corresponding section in the Materials and Methods.  Having a section on how these experiments were conducted will allow for the methods being moved from the results.

2.  There is no Figure 3D as mentioned on p. 6. line 257.   Are the bars above each column in Fig. 3 (as well as other figures) standard error or standard deviations?

3.  In Fig. 3B, what is relative mycelial diameter?  This should be explained in the Materials and Methods.

4.  P. 6, line 254, the wording should be smaller.

5.  Given that the silenced strains all grow less abundantly than the WT, how is this lack of hyphal growth taken into consideration when measuring H2O2 fluorescence or content?

6.  After each result section there is a general conclusion or suggestion of what the data mean.  Would this not be best left to the discussion section?

7.  P. 13, lines 416-417.  The genes my be involved for growth, but all strains grew.  Suggest adding ...were required for normal growth in P. ostreatus.

8.  Would the authors speculate on any reasons the PoLaeA1 gene appears to operate very differently than the other LaeA genes in this fungus.  Do other fungi have multiple LaeA genes that function differently?

Author Response

Dear Editors and reviewers:

Thank you for your letter and for the reviewers′ precious comments and suggestions concerning our manuscript entitled “Functional roles of LaeA-like genes in fungal growth, cellulase activity, and secondary metabolism in Pleurotus ostreatus” (ID is jof-1856782). These comments were all valuable and very helpful for revising and improving our paper and significantly guided our research. We have carefully revised the manuscript according to your comments. In addition, thank you for the guidelines on how to submit our revised manuscript. We hope the revised manuscript is now suitable for publication.

The sections revised according to the reviewer′s suggestions are highlighted in red text in the revised manuscript. The point-to-point replies to the reviewers′ comments are listed below.

Reviewers' comments:

Reviewer #2:
1. In the results, there is a section on the measurement of growth and biomass. However, there is no corresponding section in the Materials and Methods. Having a section on how these experiments were conducted will allow for the methods being moved from the results.
Response:
Thank you very much for your suggestion. We have added a section on the measurement of growth and biomass in the Section of “Materials and methods” accordingly (Page 3, lines 126-131).

  1. There is no Figure 3D as mentioned on p. 6. line 257. Are the bars above each column in Fig. 3 (as well as other figures) standard error or standard deviations?
    Response: Thank you very much. We have used “Figure 3C” instead of “Figure 3D” accordingly (Page 7, line 268). The bars above each column in figures are standard deviations as shown in Section of “Data analysis” in our revised manuscript.

  1. In Fig. 3B, what is relative mycelial diameter? This should be explained in the Materials and Methods.

Response: Thank you very much for your suggestion. The relative mycelial diameter = (the diameter of fungal strain)/(the diameter of the WT strain)×100%, and we have added the relative description in the Section of “Materials and Methods” accordingly (Page 3, lines 128-130).

  1. P. 6, line 254, the wording should be smaller.
    Response: Thank you very much. We have revised in our revised manuscript accordingly (Page 6, line 265).

  1. Given that the silenced strains all grow less abundantly than the WT, how is this lack of hyphal growth taken into consideration when measuring H2O2 fluorescence or content?
    Response: Thank you very much. The purpose of our experiments is to determine the differences between the silenced strains and the WT strain, therefore, the lack of hyphal growth in silenced strains is less taken into consideration when measuring H2O2 fluorescence or content. Nevertheless, the H2O2 content is measured using the mycelial pellets collected from CYM liquid medium, and the mycelial pellets of the WT and silenced strains is similar.

  1. After each result section there is a general conclusion or suggestion of what the data mean. Would this not be best left to the discussion section?
    Response: Thank you very much. We added a general conclusion of what the data mean after each result section to make readers clearer, and we believed this is more appropriate for readers to understand our paper. Thank you again.

  1. P. 13, lines 416-417. The genes may be involved for growth, but all strains grew. Suggest adding ...were required for normal growth in P. ostreatus.

Response: Thank you very much for your suggestion. We have revised the relative description accordingly (Page 13, line 437).

  1. Would the authors speculate on any reasons the PoLaeA1 gene appears to operate very differently than the other LaeA genes in this fungus. Do other fungi have multiple LaeA genes that function differently?

Response: Thank you very much for your suggestion. The PoLaeA1 gene appears to operate very differently than the other LaeA genes and we speculate that the reasons for this may be related to the different expression patterns between PoLaeA1 and other LaeA genes during developmental stages. In addition, we have added relative description in our manuscript (Page 5, line 233; Page 6, lines 234-235). Besides, there are three LaeA genes in Aspergillus luchuensis mut. kawachii that function differently on citric acid accumulation [1].

[1] Kadooka, C., et al., LaeA Controls citric acid production through regulation of the citrate exporter-encoding cexA gene in Aspergillus luchuensis mut. kawachii. Appl. Environ. Microbiol. 2020, 86, e01950-19. 10.1128/AEM.01950-19.

Thanks again for your comments and suggestions.

Best wishes,

Guang Zhang

13 Aug 2022

Reviewer 3 Report

Dear Authors,

Reviewer comments journal of fungi-1856782

The manuscript entitled „Functional roles of LaeA-like genes in fungal growth,cellulase aktivity, and secondary metabolism in Pleurotus ostreatus“ represents a useful study on sturctural and functional diversity of PoLaeA genes in Pleurotus ostreatus. Three homologs of PoLaeA genes named PoLAeA1, PoLaeA2 and PoLaeA3 were identified in Pleurotus ostreatus and their biological function was studied using the silencing constructs. Analysis of the plants containing silenced PoLaeA genes revealed that PoLaeA1 regulates intracellular polysaccharide content (IPS) by regulating intracellular ROS levels while PoLaeA2 and PoLAeA3 regulate cellulase activity via Ca2+ signalling.

The manuscript provides original novel experimental results elucidating the role of LaeA genes in the polysaccharide metabolism, ROS metabolism and calcium signalling in Pleurotus ostreatus.

However, I have a few important comments on the present manuscript which are given below:¨

1/ In Materials and methods, basic information on the statistical test and software used for the data analysis presented in the figures has to be added. In addition, basic information on the kind of statistical test used for the determination of significant differences including ANOVA and the following post-hoc test has to be added to the figure legends following the statements on the significant differences, i.e., „different lowercase letters indicate significant differences between the strains (P≤0.05) determined by ANOVA analysis, - the kind of a post-hoc multiple comparisons test has to be specified !!“.

2/ In Materials and methods, the date of access has to be added to the databases searched for the data in the study since the database content changes with time so the information on the date of access of the given database has to be added to each database cited in the study.

3/ In Figure 3A, an appropriate scale bar has to be added to the photos of fungal growth on CYM plates. In Figure 3B, an appropriate unit has to be added to the graph on mycelial diameters.

4/ The full scientific names of the fungi have to be given in the manuscript to provide important information for the readers which are not familiar with mycology. The authors should use the full fungal names, e.g., Hypsizygus marmoreus instead of H. marmoreus (line 177),  instead of  â€žA. nidulans“ (line 409) and other fungal names.

5/ The abbreviations used in the text should be explained when used for the first time, or better, a separate Abbreviations list has to be added to the manuscript. For example, in Figure 1 legend, line 209, "putative SAM binding sites“ where „SAM“ has to be explained as „S-adenosylmethionine.“

Formal comments on the etxt:

Line 180. Add the word „respectively“ at the end of the statement „The open reading frames of PoLAeA1, PoLaeA2 and PoLaeA3 are 1212, 756 and 699 bp, respectively,…“

Line 184: Add the word „respectively“ at the end of the statement „The PoLaeA1 gene encodes a 45.75-kDa protein of 403 amino acids with a pI of 4.78; poLaeA2 encodes a 28.29-kDa protein of 251 amino acids with a pI of 5.44, and PoLaeA3 encodes a 25.86 kDa protein of 232 amino acids with a pI of 4.32, respectively.“

Final recommendation: Accept after a minor revision.

Author Response

Dear Editors and reviewers:

Thank you for your letter and for the reviewers′ precious comments and suggestions concerning our manuscript entitled “Functional roles of LaeA-like genes in fungal growth, cellulase activity, and secondary metabolism in Pleurotus ostreatus” (ID is jof-1856782). These comments were all valuable and very helpful for revising and improving our paper and significantly guided our research. We have carefully revised the manuscript according to your comments. In addition, thank you for the guidelines on how to submit our revised manuscript. We hope the revised manuscript is now suitable for publication.

The sections revised according to the reviewer′s suggestions are highlighted in red text in the revised manuscript. The point-to-point replies to the reviewers′ comments are listed below.

Reviewers' comments:

Reviewer #3:
The manuscript entitled “Functional roles of LaeA-like genes in fungal growth,cellulase aktivity, and secondary metabolism in Pleurotus ostreatus” represents a useful study on sturctural and functional diversity of PoLaeA genes in Pleurotus ostreatus. Three homologs of PoLaeA genes named PoLAeA1, PoLaeA2 and PoLaeA3 were identified in Pleurotus ostreatus and their biological function was studied using the silencing constructs. Analysis of the plants containing silenced PoLaeA genes revealed that PoLaeA1 regulates intracellular polysaccharide content (IPS) by regulating intracellular ROS levels while PoLaeA2 and PoLAeA3 regulate cellulase activity via Ca2+ signalling.

The manuscript provides original novel experimental results elucidating the role of LaeA genes in the polysaccharide metabolism, ROS metabolism and calcium signalling in Pleurotus ostreatus.

However, I have a few important comments on the present manuscript which are given below:¨

  1. In Materials and methods, basic information on the statistical test and software used for the data analysis presented in the figures has to be added. In addition, basic information on the kind of statistical test used for the determination of significant differences including ANOVA and the following post-hoc test has to be added to the figure legends following the statements on the significant differences, i.e., „different lowercase letters indicate significant differences between the strains (P≤0.05) determined by ANOVA analysis, - the kind of a post-hoc multiple comparisons test has to be specified !!“.
    Response: Thank you very much for your suggestion. We have added basic information on the statistical test and software used for the data analysis in the Section of “Data analysis” (Page 4, lines 187-190). In addition, we have added the basic information on the kind of statistical test used for the determination of significant differences in the Figure legends (Page 6, lines 245-246; Page 7, line 274; Page 8, lines 309; Page 9, line 318; Page 10, lines 344-345; Page 11, line 378; Page 12, lines 413-414).

  2. In Materials and methods, the date of access has to be added to the databases searched for the data in the study since the database content changes with time so the information on the date of access of the given database has to be added to each database cited in the study.
    Response: Thank you very much for your suggestion. We have added the date of access to the databases for the data in the revised manuscript accordingly (Page 2, lines 95 and 98; Page 3, lines 103, 106-107 and 112; Page 4, line 190; Page 5, lines 214-217).

  1. In Figure 3A, an appropriate scale bar has to be added to the photos of fungal growth on CYM plates. In Figure 3B, an appropriate unit has to be added to the graph on mycelial diameters.

Response: Thank you very much for your suggestion. We have added an appropriate scale bar into Fig. 3A (Page 7, line 270) and added an appropriate unit into Figure 3B (Page 7, line 270) accordingly.

  1. The full scientific names of the fungi have to be given in the manuscript to provide important information for the readers which are not familiar with mycology. The authors should use the full fungal names, e.g., Hypsizygus marmoreus instead of H. marmoreus (line 177), instead of, A. nidulans (line 409) and other fungal names.
    Response: Thank you very much for your suggestion. We have given the full scientific names of the fungi in the revised manuscript (Page 5, line 205 and 211; Page 13, lines 429, 438, 447-448, 453, 459 461, and 466).

  1. The abbreviations used in the text should be explained when used for the first time, or better, a separate Abbreviations list has to be added to the manuscript. For example, in Figure 1 legend, line 209, "putative SAM binding sites “where SAM” has to be explained as, S-adenosylmethionine.”
    Response: Thank you very much for your suggestion. we have added the explain for the abbreviations used in the text accordingly (Page 5, line 203 and 218; Page 7, line 294, Page 8, line 295, Page 11, lines 360-361).

  1. Line 180. Add the word, respectively “at the end of the statement, The open reading frames of PoLaeA1, PoLaeA2 and PoLaeA3 are 1212, 756, and 699 bp, respectively.”

Response: Thank you very much for your suggestion. We have revised accordingly (Page 5, line 197).

  1. Line 184: Add the word, respectively “at the end of the statement, The PoLaeA1 gene encodes a 45.75-kDa protein of 403 amino acids with a pI of 4.78; poLaeA2 encodes a 28.29-kDa protein of 251 amino acids with a pI of 5.44, and PoLaeA3 encodes a 25.86 kDa protein of 232 amino acids with a pI of 4.32, respectively.”

Response: Thank you very much for your suggestion. We have revised accordingly (Page 5, line 200).

Thanks again for your comments and suggestions.

Best wishes,

Guang Zhang

13 Aug 2022